# Personal Exposure to Black Carbon, Particulate Matter and Nitrogen Dioxide in the Paris Region Measured by Portable Sensors Worn by Volunteers

**DOI:** 10.3390/toxics10010033

**Published:** 2022-01-11

**Authors:** Baptiste Languille, Valérie Gros, Bonnaire Nicolas, Cécile Honoré, Anne Kaufmann, Karine Zeitouni

**Affiliations:** 1Department of Chemistry, University of Oslo, 0315 Oslo, Norway; baptiste.languille@kjemi.uio.no; 2Laboratoire des Sciences du Climat et de L’environnement UMR1572, IPSL/CEA/CNRS/UVSQ, L’Orme des Merisiers, CEA Saclay, CEDEX, 91191 Gif sur Yvette, France; nicolas.bonnaire@lsce.ipsl.fr; 3Airparif, 75004 Paris, France; cecile.honore@paris.fr (C.H.); anne.kauffmann@airparif.fr (A.K.); 4Mairie de Paris, Direction de la Voirie et des Déplacements, 75013 Paris, France; 5Laboratoire DAVID, Université de Versailles Saint-Quentin-en-Yvelines, 78035 Versailles, France; karine.zeitouni@uvsq.fr

**Keywords:** mobile measurements, ambient monitoring, urban pollution, microenvironments, air quality, Île-de-France

## Abstract

Portable sensors have emerged as a promising solution for personal exposure (PE) measurement. For the first time in Île-de-France, PE to black carbon (BC), particulate matter (PM), and nitrogen dioxide (NO_2_) was quantified based on three field campaigns involving 37 volunteers from the general public wearing the sensors all day long for a week. This successful deployment demonstrated its ability to quantify PE on a large scale, in various environments (from dense urban to suburban, indoor and outdoor) and in all seasons. The impact of the visited environments was investigated. The proximity to road traffic (for BC and NO_2_), as well as cooking activities and tobacco smoke (for PM), made significant contributions to total exposure (up to 34%, 26%, and 44%, respectively), even though the time spent in these environments was short. Finally, even if ambient outdoor levels played a role in PE, the prominent impact of the different environments suggests that traditional ambient monitoring stations is not a proper surrogate for PE quantification.

## 1. Introduction

Studies based on small sensors for air pollutant measurements are associated with very different objectives and fields: regulatory monitoring [1], the use of mapping and models [2], monitoring of industrial sites [3], the characterisation of very specific environments [4] and sources [5], the measurement of health impacts [6], and even studies of climate concern [7].

The quantification of personal exposure (PE) to pollutants is a field of particular interest within the sensors community, even though there is no clear consensus on the definition of the term itself nor standard procedure for its quantification. Indeed, PE has been employed to define slightly different metrics. For instance, Ostro et al. [8] deduced PE from chemistry transport models (and, therefore, only considered outdoors pollutants). PE was also measured on the field, either using static measurement [9] or mobile sensors [10]. It is also possible to combine pollution sensors with respiratory frequency and flow measurements to calculate exposure as the amount of pollutants actually inhaled [11,12,13]; more advanced research has even studied the impact of the breathing cycle on the accuracy and representativeness of the sampled air [14].

Several studies have already focused on PE quantification. However, many suffered from poor temporal resolution, such as those using passive tubes for nitrogen dioxide (NO_2_) quantification or filter-based measurements for particulate matter (PM) [15,16,17,18]. The paramount importance of the sampling time resolution has already been recognized [19]. Other works focused on very specific activity or groups of persons, such as road traffic [12,20] or children [18,21,22,23]. On the other hand, the number of volunteers carrying the sensors and the duration of the campaigns was sometimes low: three volunteers for 2-h trips [12], or only one volunteer for successive 24-h measurements [24], or only a proof-of-concept of a few hours [25]. These limitations may preclude the characterization of PE associated with different environments (considered individually). Even among studies exceeding the above-mentioned limits [26,27,28,29], few have conducted investigations on several pollutants simultaneously and even fewer have included measurements of both gases and particles.

Portable sensors are powerful tools for innovative investigations. Because they are less expensive than traditional instruments, as well as small and lightweight, many units can be purchased and as many people can be easily equipped for 24/7 personal exposure monitoring. Nevertheless, the questionable accuracy and precision of such sensors as well as the drawbacks inherent to citizen science are limitations that need to be considered. Different strategies, with a degree of correction, have been implemented. Some works [13] rely only on following the known procedures (manufacturer manual and statistical treatment), other studies assess the sensors versus reference instruments [30,31], and others even apply a correction on the sensors [32]. Regarding citizens science, several published works document the best practices and errors to be avoided [22,23,33].

Île-de-France (the region around Paris) is the most populated region in France (~12 million inhabitants). Its airborne pollutants exceed thresholds and WHO recommendation limits; NO_2_, PM_2.5_, PM_10_, and ozone, in particular, are still a concern [34]. Studies have already investigated PE in Île-de-France; nevertheless, this research focused on very specific volunteers or distinct occupations, including taxi drivers [35], office workers [36], particular activities (such as cyclists) [37], or the narrow population category (such as children) [21]. However, investigations at the individual level which aim to characterise PE of the general public in Île-de-France remain scarce.

The Polluscope project (funded by the French National Research Agency) precisely addresses the previously mentioned limitations and suggests a robust experimental protocol based on optimized previous works, including a complete characterisation of the sensor performances [30]. This project initiated in 2016 aims at quantifying PE for the general public in Île-de-France. To achieve this goal, five measurement campaigns were conducted in the frame of the project and two are scheduled for the years 2021–2022. These campaigns (conducted in different seasons) involved a total of 139 volunteers equipped with portable and geolocalised sensor units all day long during one week. These large experimental deployments yielded a 1-min data set for PM, black carbon (BC) and NO_2_.

In this paper, PE is considered for a given pollutant as the concentration to which an individual is exposed to. We will discern between overall exposure (over one or several days) and PE relative to specific environments. The present work is based on the two first campaigns of the project plus another field deployment conducted outside Polluscope. This paper aimed at (i) giving a first quantification of PE for BC, NO_2_, and PM in Île-de-France, (ii) defining each environment contribution; and (iv) discussing the relevance of using ambient measurement as a PE surrogate.

## 2. Materials and Methods

The experimental strategy for PE measurement went through the deployment of three sensors. Three campaigns were conducted, where volunteers were equipped with several units of each sensor for a week.

### 2.1. Characterization of the Sensors

The characterization of the sensors was conducted according to a robust and validated protocol, as extensively detailed by Languille et al. [30]. In 2017, a selection stage was conducted; it consisted in both static and mobility tests, based on the sensors available on the market at that time. To begin with, all the sensors of interest were put in measurements for several continuous days in ambient air, sheltered from rain. The co-located reference instruments, maintained by the SIRTA station (Available online: http://sirta.ipsl.fr (accessed on 5 January 2022)) and included in the research infrastructure ACTRIS (Available online: https://www.actris.eu/ (accessed on 5 January 2022)), allowed a first appraisal of the sensors’ quality and helped to rule out sensors which gave aberrant signals. A short list was then drawn up. In order to confirm the ability of these sensors to perform PE measurements, they were submitted to mobility tests. Three people were equipped with three units of each sensor’s model. During the day, they followed a previously set route, going through different environments and comprising transient stops close to measuring stations taken care of by Airparif (the accredited association for air quality monitoring in Île-de-France, Available online: https://www.airparif.fr (accessed on 5 January 2022)). From this selection step, three sensors were selected out of the eight tested devices: AE51 (AethLabs, San Francisco, CA, USA) for BC measurements; Canarin (developed within UMR 7606 Sorbonne University—CNRS) for PM measurements; and Cairsens (Envea, Poissy, France), previously called Cairsens, for NO_2_ measurements.

Then, the selected sensors underwent the assessment stage; reproducibility between about fifteen units was quantified, as well as potential interferences with humidity and other pollutants. In 2018, the fifteen units of each sensor were tested simultaneously in ambient air, in static measurements, for a week. For the AE51 and the Canarin sensors, references were provided by the colocation of the SIRTA instruments. In order to encounter high enough NO_2_ levels, the Cairsens sensors were tested in the direct vicinity of a traffic-oriented Airparif station, used as reference. In Languille et al. [30] and its Appendix A, the Cairsens sensors demonstrated their ability to capture a coarse NO_2_ concentration variability, even for ambient levels close or below 20 ppb. Due to a more complex setup, only one campaign (from 28 August to 4 September 2018) was conducted for the Cairsens, at the traffic site, whereas two campaigns (from 26 June to 2 July, and from 5 to 9 November 2018, respectively) were organised at SIRTA for the AE51 and Canarin sensors. These qualification campaigns were conducted as close as possible to the PE measurements and, therefore, accounted for any eventual drift endured by the sensors over the weeks and months.

The integrated performance index (IPI) designed by Fishbain et al. [38] was calculated to evaluate the reliability of pollutant sensors. This metric combines seven parameters: root mean squared error (RMSE); Pearson, Kendall, and Spearman correlations; data loss; match score (metric assessing the ranking order similarities); and the lower frequencies energy (LFE, estimating the sensor ability to properly capture the signal variability). The resulting IPI ranged from 0 (the worse) to 1 (the best grade).

Generally, all the sensors gave satisfactory results. The correlations with the reference instruments were elevated (mean r = 0.8); the data loss was very seldom (less than 10% on average) and the RMSE was low (a third on average) compared to the ambient levels (measured by the reference instruments). More specifically, the AE51 sensors were graded a high IPI (around 0.8), an averaged Pearson correlation of 0.8, and a reasonably low RMSE (below 350 ng·m^−3^). The Cairsens sensors showed a Pearson correlation (r) above 0.75 and a RMSE always below the quantification limit given by the manufacturer (20 ppb), resulting in a satisfactory mean IPI of 0.77. The Canarin sensors showed both a high Pearson correlation (mean r = 0.84), a low RMSE (6 µg·m^−3^ on average), and a high mean IPI (0.74). It is noteworthy that the PM sensor extracts three PM size classes from one single measurement. The unexpectedly high correlation between the PM size classes suggests that the Canarin was not able to detect all the specificities belonging to each size class. In the following, this study focuses mainly on PM_2.5_ as it was both the best measured [30] and the most relevant class regarding epidemiological studies [39]. The detailed performances for all the BC, NO_2_, and PM sensor are given in Appendix A).

Finally, the controlled chamber tests helped to pinpoint that the AE51 and Cairsens sensors could show an artefact in case of a sudden and pronounced humidity variations. This is an important finding that is to be taken into account for the quality control step.

### 2.2. Presentation of the Three Measurement Campaigns

All the three devices were deployed during field campaigns, where each volunteer from the general public was equipped with one Cairsens and one Canarin. Due to its higher cost, only six units of the AE51 were available and distributed among the volunteers. They were asked to wear them during the day and to put them on a relevant place when the persons are at home (close to their bed when they are sleeping for instance). In total, 37 volunteers were involved, which represents 65 days of field campaign and 1,144,616 data points (at 1 min time step).

The three campaigns are hereinafter called spring, autumn, and winter campaigns. The first two were included in the Polluscope project, and all the sensors were shared between the volunteers and the measurements, therefore, took place in parallel. On the contrary, the winter campaign was a side experiment, with limited manpower and only one set of sensors (one unit of each). This third campaign was included in another project; with a focus on wood burning, the involved volunteers were all equipped with wood burning appliances at home. Even considering these specificities, all three experiments used the same three sensors and the Île-de-France region as geographical location. Table 1 summarizes the three campaign dimensions. The volunteer names were coded with a two-letter tag (AA, AB, AC, and so on until BI) for anonymization purposes. The association between the sensors and the volunteers is presented in Appendix A).

### 2.3. Data Invalidation

These portable instruments were submitted to harsh conditions, which induced significant artefacts. Languille et al. [30] showed the impact of environmental changes. An in-depth data curation was conducted manually. More precisely, abrupt changes in humidity, temperature, or atmospheric pressure (monitored by the sensors) were tracked. Single measurements with clearly unexpected value (obvious outliers exceeding the neighbour values by one order of magnitude), even without simultaneous environmental change recorded, were invalidated (because most likely due to an artefact). As a result, several data points in the range of a few thousand were invalidated, which represents about 1% of the total measurements.

The limits of quantification are not specified by the manufacturer for the AE51 and Canarin. In order to keep all the variability information, as well as the same post processing of the data from all three sensors, no data were ruled out because of low value.

## 3. Results

### 3.1. Overview of the Three Campaign Results and Comparison with Other Studies

The two campaigns conducted in the frame of the Polluscope project (namely spring and autumn) were achieved with the same protocol and the same process for the recruitment of volunteers. Figure 1 presents the volunteer positions recorded by the embedded GPS (1 min data points); a limitation of this approach is the weak signal received in certain environments (especially in the underground) leading to sporadic lack of location data. The routes followed by the volunteers during those two campaigns were analogous, with travels both in the inner Paris city and in the suburbs. Conversely, the volunteers’ recruitment for the winter campaign (outside the Polluscope project) followed a different process, which resulted in noticeable different routes followed by the volunteers. As visible in Figure 1, in winter, the volunteers commuting were oriented towards the suburban region and slightly less in the very centre of the Parisian metropolis. This must be taken into account when comparing the different campaign results. Indeed, during the spring and autumn campaigns, the volunteers were more likely to be in close contact to heavy traffic and other sources that characterise highly densely populated areas.

Table 2 gathers the PE measured during the three campaigns; more precisely, the mean, standard deviation, as well as the maximum hourly mean, for BC (in ng·m^−3^), NO_2_ (in ppb), and PM (in µg·m^−3^). On average, BC and PM concentrations were higher (in the range of +50% and +200% respectively) in autumn compared to spring. This difference could be attributed to seasonal variations in the ambient concentrations, as it was observed before [40]. However, during the winter campaign, PE to BC and PM were lower than in autumn, and even lower than in spring for BC. This might be due to the significant difference in the dwelling and working locations of the volunteers. Indeed, suburban areas are known to be less impacted by PM and BC pollution [34,41]. Nevertheless, it is worth noticing that high hourly maxima were measured during the winter campaign, which is most likely due to the proximity to wood burning appliances. On the opposite, the lowest hourly maximum was observed for NO_2_ during the winter campaign. As the high hourly maximum is linked with traffic proximity, this assessment highlights the lesser importance of traffic in PE during the winter campaign (which is not linked with the season, but due to the volunteers location, as mentioned above). These observations underline the preponderance of the activities as PE drivers. It is noteworthy that no clear difference in the mean was observed for NO_2_ over the three campaigns. The high limit of detection of the Cairsens sensor is probably the reason [30]. It is, therefore, more relevant to study this sensor’s data during periods of higher concentration or focusing of the variations rather than general means.

An abundant literature studying PE measurement using small sensors is available. Nevertheless, a large part of these studies focus on epidemiology. Furthermore, NO_2_ is, most of the time, measured with passive samplers which induce both a coarse time resolution and a high limit of detection. Table 2 presents a selection of the most relevant studies conducted worldwide to be compared with the PEs measured in this work in Île-de-France. All the discussed studies used the AE51 for BC measurements, the Cairsens was only used by Borghi et al. [42], and the Canarin was not used at all due to its newer release.

Large differences were observed in measured levels among these studies. In Stockholm, Merritt et al. [12] observed slightly higher levels than in our study, as it was expected in traffic-influenced environments. In Brisbane and Birmingham, relatively low PE to BC were published—600 ng·m^−3^ and 1300 ng·m^−3^, respectively—which is still in the same range as PE measured in Île-de-France [24,31]. In Delhi, and especially in winter, PE to PM and BC were more than one order of magnitude higher than our findings [43]. This significant discrepancy is probably mostly due to the high ambient levels monitored is this city [44]. Regarding PM exposures, important discrepancies were observed between western cities, such as Birmingham, Milan, or Göteborg (in the single- or low double-digit µg∙m^−3^ range), as well as heavily polluted cities, such as Delhi and Beijing, with the mean PE exceeding hundred µg·m^−3^ [43,45]. Concerning NO_2_, results from all studies ranged from single-digit to low double-digit ppb [17,18,31,42,46]. This is consistent with our results, taking into account the high Cairsens detection limit (20 ppb).

To conclude, the overall PE values measured in Île-de-France in this present study are much lower than results from heavily polluted Indian or Chinese metropolis (hourly maxima in Paris were in the same range as the mean in Delhi). The results disclosed here are closer to studies conducted in more similar western cities.

Such studies involving volunteers equipped with sensors allow comparisons between each other, which is valuable to understand the height of the exposure range that citizens are exposed to. However, one of the major assets of the portable sensors is the PE characterisation for each single volunteer with a fine time resolution.

Figure 2 shows an exemplary time series of the three sensors carried by volunteer AH; it shows the typical signal variation observed during the campaigns. Both BC, NO_2_, and PM measurements presented long low-value periods and short events with high concentrations. More specifically, first assumptions can be made based on these time series, without considering the environment tags (identified based on the schedules given by the volunteers studied in the next section). For instance, the two first BC spikes that occurred on 19 and 20 June in the late afternoon were concomitant with NO_2_ peaks. The time in the day and the correlation of these two pollutants suggest that these were traffic episodes. On the contrary, in the evening of 19 June (around 22:00), another episode was characterised by a chemical fingerprint only driven by a huge PM signal (BC and NO_2_ stayed low). This event resembled an indoor pollution episode. Indeed, long tailing after the peak could be due to non-dispersive conditions peculiar to indoor environments. Furthermore, the chemical fingerprint of this episode rules out any combustion source (such as traffic) and is more alike cooking or cleaning activity patterns. Nevertheless, these environment and activity attributions were based on assumptions and need to be confirmed for a strengthened identification.

### 3.2. Environments: Characterization and Contribution to PE

Volunteers filled in detailed schedules about their different activities throughout the day. This valuable extra information allowed investigating PE for each single environment. Variations in level of details, completeness, and accuracy in the schedules given by volunteers led us to distinguish five different environment types, namely “commuting” (car, motorbike, bus, train, underground, etc.), “outdoor” (for outdoor activities other than commuting), “indoor” (by default for time spent indoor), “polluted indoor” (for obvious polluted indoor environments or activities such as cooking, hoovering, smoking, etc.), and “wood burning” (nearby a running wood burning appliance, specific to the winter campaign). It is noteworthy that no polluted indoor episode was identified during a “wood burning” period. The environments identified using the volunteer’s schedule are plotted on Figure 2. The most frequently visited environment was indoor, where the levels were low and stable. More seldom, polluted indoor episodes were observed, as shown on the time series, on 19 June around 22:00, which was due to cooking activities. During commuting periods of time (early morning and afternoon), increased PE was monitored, especially for BC and NO_2_. Like most volunteers, time spent outdoors is short. This is an expected result knowing that, in France, more than 85% of the lifetime is spent indoors [47]. It is nevertheless possible to pinpoint transient outdoor periods on Figure 2 (after the evening commuting trips and before the morning one). This first overview helped to show the huge impact that the activities had on PE. Nevertheless, this qualitative approach is not sufficient.

Table 3 summarizes the mean PE by environment for each campaign. For all pollutants, the lowest levels were measured indoor. On the opposite, the highest BC and NO_2_ PE were observed during commuting. Polluted indoor environments were characterized by the highest PM_2.5_ levels and by rather high BC values (especially close to wood burning appliances). At last, outdoor, varied levels were monitored, depending on the conducted activities and possibly the ambient levels.

Studying the mean PE encountered in each environment offers valuable information. However, to fully understand the possible influence that this represents to health, it is of great importance to include the amount of time spent in each environment. The obtained metric is the contribution of each environment to the total PE. Its formula is given below for a given pollutant.
$$Contribution_{Env.\;\;i}=\frac{Exposure_{Env.\;\;i}}{Exposure_{Total}}\times \%Time_{Env.\;\;i}$$

A systematic data processing was conducted to determine the PE contribution of each environment for a given pollutant, using this formula. The resulting contribution (*Contribution_Env. i_*) is the ratio between the mean exposure in this environment and the total exposure (i.e., for all the environments) multiplied by the percentage of time spent in this particular environment. The figures for all volunteers are presented in Appendix A), and two symptomatic examples are shown here (Figure 3 and Figure 4).

Volunteers AI and AH spent most of the time indoor (from 78% to 93%), which is in agreement with previous studies [48]. Nevertheless, as the concentrations measured indoors are rather low, its contribution is lower than the time spent in it. On the contrary, the few and short indoor polluted events were characterized by such high values that can represent a significant contribution to the total PE. This is clearly visible in Figure 3, which shows how the volunteer AI spent only 2% of its time in polluted indoor environments, but this contributed to 41% of its PE to PM_2.5_. Similarly, little time was spent in commuting, but this environment can represent an important contribution to BC and NO_2_ PE (Figure 4: 6% of the time but 34% and 15% of the PE to BC and NO_2_, respectively).

Epidemiological studies showed that both acute and long-term exposures have severe adverse health effects [49]. This underlines the importance of monitoring both the ambient levels and the short-term exposures associated with specific activities or environments.

The strong impact of short time spent in polluted environments is obvious. On the other hand, PE was higher during autumn for most environments, in particular outdoor, which follows the ambient measurements trend. This would suggest that ambient levels could influence PE both outdoor and in the other environments. Based on that hypothesis, several studies proposed using ambient monitoring as a surrogate for PE quantification. This approach is discussed in the next section.

### 3.3. Temporal Variability of PE and Correlation with Urban Background Levels

Computing diurnal cycles of compounds is a commonly used methodology for source apportionment [50]. Therefore, we analysed the diurnal patterns of the measured pollutants. Even if every single individual had his own schedule, common time habits were observed. Figure 5 presents the average normalised diurnal pattern (normalisation achieved by dividing by the mean value) for all the volunteers for the three campaigns. The BC pattern showed two peaks: the first one occurred in the morning (around 08:00) was rather short, while the second one started in the late afternoon (around 17:00) and lasted late, slowly decreasing overnight. PM showed higher values overnight (from 20:00 to 02:00 roughly), with a long peak beginning after the evening BC one. NO_2_ had a clearly different cycle with maximum values during the day (from 08:00 to 21:00 approximately) and very low levels at night. These diurnal patterns resemble the characteristic daily patterns attributed to specific source in Île-de-France [50]. More precisely, the traffic pattern showed two peaks, and the domestic wood burning pattern was defined by a longer and later peak in the evening. These diurnal patterns are similar to the BC and PM_2.5_ diurnal cycles displayed in Figure 5. Nevertheless, it is not straightforward to attribute this pattern to ambient outdoor levels variations or to the volunteer’s activities (and therefore the visited environments) following a certain schedule. These two possible explanations are discussed below.

Intuitively, it seems that PE is both driven by outside pollutant levels and concentrations encountered in specific environment, as well as potentially indoor levels. A critical question in the epidemiological field is whether ambient outdoor monitoring conducted in background stations is suitable as a surrogate for PE quantification. Based on satisfactory enough high correlation between fixed outdoor monitoring stations and PE measurements, several studies came to the conclusion that ambient levels have a strong impact on PE and, subsequently, that ambient monitoring can be used as a proper surrogate for PE quantification [11,15,16,43,45,51].

More specifically, Kim et al. [16] showed a correlation of 0.69 for PM_2.5_ (median of the 28 volunteers’ Spearman coefficients between ambient monitoring and their PE). Regarding PM_10_, Hsu et al. [51] observed Spearman correlation coefficients (PE versus ambient measurements) ranging from −0.18 to 0.79 among the 24 participants. Du et al. [45] published Spearman coefficients of 0.81 for PM_2.5_ (PE versus fixed monitoring site, N = 114). Similarly, Johannesson et al. [15] presented a correlation of 0.61 for PM_2.5_ (N = 25). Regarding BC, Pant et al. [43] observed a Spearman correlation of 0.72 (N = 204) and 0.28 in winter and summer, respectively (PE to BC versus PM_2.5_ ambient measurements). Johannesson et al. [15] presented similar correlations using black smoke (BS) as a proxy for BC ambient measurements (r = 0.65). Du et al. [45] published Spearman coefficients of 0.61 for BC (PE versus fixed monitoring site, N = 114). Obtaining these results, all the above-mentioned studies concluded that fixed ambient monitoring stations could be used as appropriate surrogates for PE quantification.

However, it seems noteworthy to highlight some limitations of this reasoning. First, the Spearman correlation was considered in all these works (even if the normality of the data sets are not systematically discussed). Unlike the Pearson correlation, it does not quantify the linear relation between the two datasets, but the ranking order consistency of the elements. For instance, Spearman correlation would not detect some disagreements between ambient and PE measurements, such as discrepancies for extreme values. In the previous sections, we showed that portable sensors were able to monitor quick environment changes. This fine temporal resolution was possible because of the high measurement frequency (time step of one minute). However, the correlation studies mentioned above considered hourly [43] or daily averages [15,16,45,51]. Increasing the averaging time is expected to artificially increase the correlation. Lastly, the representativeness of the sample size was sometimes questionable (only six volunteers in winter in [43] and a total number of only 25 data points in [15]).

For comparison purposes, similar correlation calculations were conducted in the present work and, since the previously mentioned studies considered the whole available PE data set (including indoors), it was necessary to apply the same calculation protocol for a relevant comparison. Comprehensive correlations were computed for both Spearman and Pearson, whereby each single volunteer was considered separately, and each campaign was considered as a whole for time resolutions of 1 h and 1 day, as well as for the three measured pollutants (BC, NO_2_, and PM_2.5_). These comprehensive correlation results are presented in Table 4. The considered ambient monitoring stations were: Paris XIII^e^ for BC measurements; Paris VII^e^, Paris XII^e^, and Paris XIII^e^ for NO_2_ measurements (three stations were available); and Paris IV^e^ for PM_2.5_. These stations are located on the map presented in Figure 1.

The first observation is that correlations out of the 1-day average data set are higher than the 1-h average (about two thirds of the times). This means that the daily averaging artificially increases the correlation. Regarding the correlations among each volunteer considered separately, very wide ranges were observed (columns “In-vol” in Table 4 for the value ranges and detailed results in Appendix A), particularly for NO_2_ where correlations often went from extreme negative to extreme positive. This is partially explained by the few amount of available data points, even leading to sporadic perfect correlations (−1.00 or 1.00) when only two points could have been computed. This highlights the primary importance of working with data sets as solid and substantial as possible, and being cautious with less exhaustive ones.

In spite of these limitations, this correlation study showed a few clear trends. It is notable that BC and, to a lesser extent, PM_2.5_ are the pollutants that show the highest correlations. These two pollutants generally showed higher correlations during the autumn campaign compared to spring. In comparison with others studies, fairly high correlations were obtained. For instance, regarding BC, Johannesson et al. [15] and Du et al. [45] found lower correlations than in this work for the autumn campaign (for 1-day average: 0.65 and 0.61, respectively, compared to r_S_ = 0.74).

However, the correlation results were highly variable, in term of seasons, pollutants, calculation method, average period of time, and among the volunteers. In addition, in Section 3.2, we demonstrated the huge impact of activities and visited environments on each single volunteer’s PE. This predominant variability in PE among individuals confirms the primary importance to measure PE at the person-to-person level. This questions the relevance of using fixed monitoring stations as surrogates for PE quantification.

## 4. Discussion and Conclusions

Environment contributions were successfully quantified, but only a relatively small amount of volunteers filled in their schedules in a comprehensive way. Larger campaigns (with more diversity among volunteers), additional measured pollutants, and possibly new tools to track the volunteer activities (such as tablets) would be efficient ways to improve the representativeness of the results. More consistency among the volunteers from one campaign to another (keeping the same volunteers) would strengthen the comparison consistency in-between seasons. The high limit of detection of the Cairsens sensor is another limitation of this study. Consequently, conclusions regarding NO_2_, especially in low-level environments (such as indoor), are not as solid as for the other pollutants. A more accurate NO_2_ sensor would solve this drawback. The Cairsens manufacturers improved the limit of detection, which is now below 10 ppb [52]. It is also noteworthy that PM sensors might suffer from questionable performances in specific environments. Indeed, it was recently shown that Canarin sensors significantly underestimate PM where the chemical composition is significantly different from outdoor air, such as underground stations. This is due to the difference of the average particles density (a parameter used to convert the number of particles as determined by the optical sensor into PM concentration) in underground stations, which are highly loaded with metals (Petit et al., unpublished data). This could explain why, as seen in Figure 2, the BC values were higher (up to 30 µg·m^−3^) than the PM values (below 20 µg·m^−3^) during the first commuting episode. Regarding the correlation calculations between PE measurements and ambient monitoring, four background stations were used. Another perspective would be to fetch the ambient data from the closest available station or even from the output of the predictive model used by Airparif (combination of ESMERALDA, Heaven, and ADMS-Urban models) [53].

Despite its limitations, this study yields valuables conclusions. This was the first time that PE to BC, NO_2_, and PM was measured in Île-de-France using portable sensors carried by volunteers. Results showed similar levels as those previously measured in comparable western urban environments, but much lower than in heavily polluted Indian or Chinese metropolis.

Environment contributions were successfully quantified, showing that some activities can represent significant parts of the total PE, even with a short time spent in these environments, especially in commuting trips (for BC and NO_2_), polluted indoor (for PM), and wood burning vicinity (BC and PM). Taking actions on these environments is, therefore, the most efficient way to mitigate PE.

Higher levels were observed in autumn than in spring, and lower levels were observed in a more suburban-oriented campaign than in the more densely populated area. This shows the importance of both seasonal and geographical gradients.

At last, we investigated a remaining open question of particular interest in epidemiology, i.e., whether fixed ambient monitoring stations act as proper surrogates for PE quantification. Rather high correlations were observed between ambient levels and PE measurements (e.g., r_S_ = 0.74 for 1-day average BC during the autumn campaign). Nevertheless, the varied observed correlations (PE versus ambient monitoring) in between volunteers, as well as for different pollutants, seasons, campaigns, averaging time periods, and monitoring stations, demonstrated the complexity and the diversity of the PE among each individuals. This underlines the questionable use of ambient monitoring as a proper PE surrogate and highlights the need to keep refining PE quantification, possibly using portable sensors, potentially associated with results from multiple static instruments localised in various environments or high-resolution modelling tools.

## Figures and Tables

**Figure 1 toxics-10-00033-f001:**
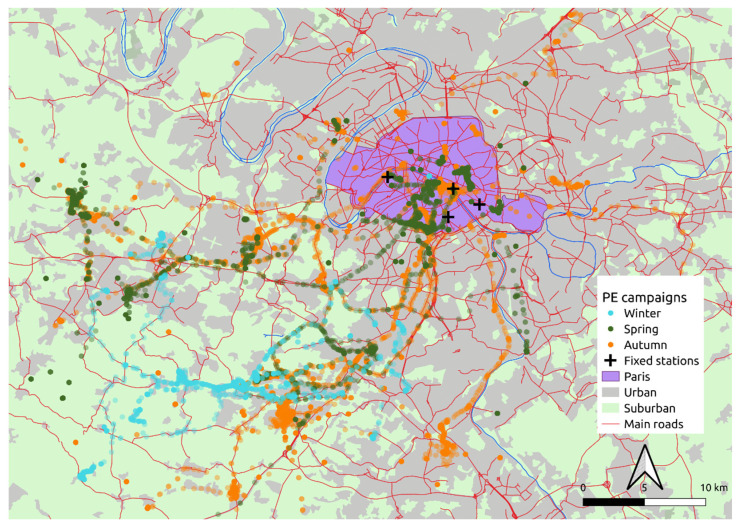
Location of volunteer measurements during the three campaigns over the Île-de-France region as well as the monitoring stations used in this study.

**Figure 2 toxics-10-00033-f002:**
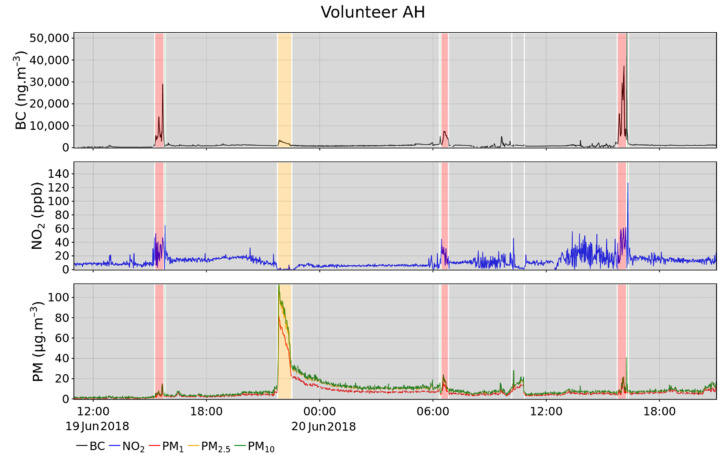
Exemplary time series of the measured PE to BC, NO_2_, and PM over four days (volunteer AH, spring campaign).

**Figure 3 toxics-10-00033-f003:**
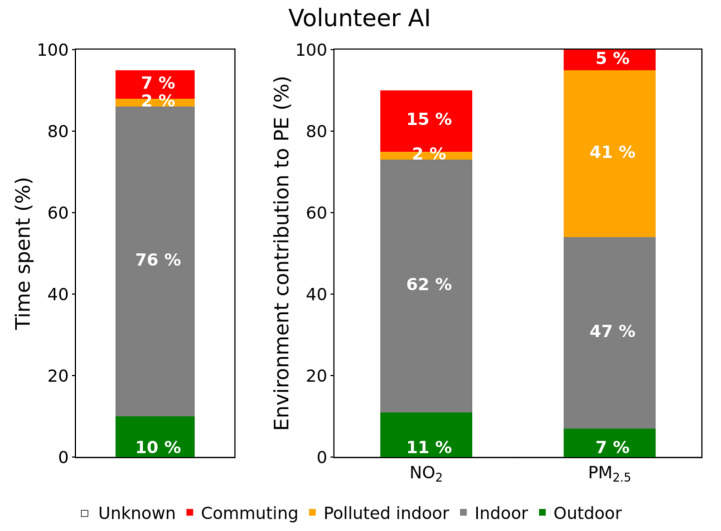
Time spent in each visited environment (left hand bar) and environment contribution to the total PE (volunteer AI, spring campaign).

**Figure 4 toxics-10-00033-f004:**
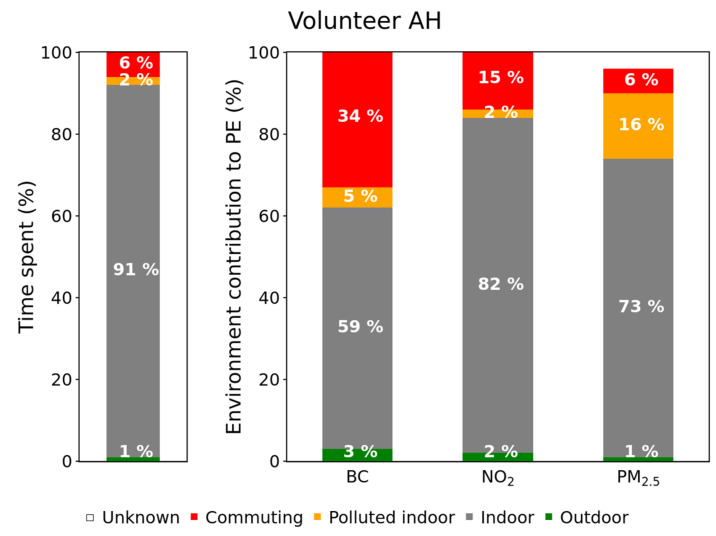
Time spent in each visited environment (left hand bar) and environment contribution to the total PE (volunteer AH, spring campaign).

**Figure 5 toxics-10-00033-f005:**
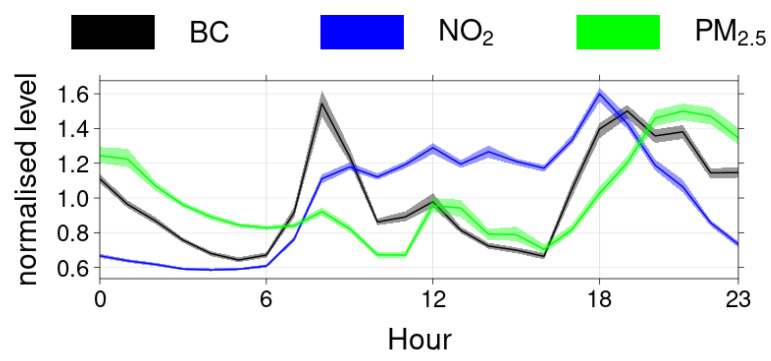
Normalised diurnal cycle (divided by the mean value) for all volunteers and for the three campaigns for BC, NO_2_, and PM_2.5_, respectively.

**Table 1 toxics-10-00033-t001:** Overview information on the three campaigns (time step = 1 min).

Campaign	Start–End	Number of Volunteers	Number of Measurements
Spring	18–22 June 2018	16	302,506
Autumn	19–26 November 2018	15	573,505
Winter	15 January–17 March 2019	6	268,605

**Table 2 toxics-10-00033-t002:** Mean measured exposure and standard deviation (sd) in different studies (BC in µg∙m^−3^, NO_2_ in ppb and PM in µg∙m^−3^).

	Study	BC	NO_2_	PM_1_	PM_2.5_	PM_10_
This study	Spring mean (sd)	1.04 (1.97)	9 (10)	5 (10)	7 (16)	8 (17)
	Spring hourly max.	7.49	45		311	
	Autumn mean (sd)	1.68 (1.52)	9 (12)	15 (18)	22 (32)	24 (36)
	Autumn hourly max.	8.97	172		490	
	Winter mean (sd)	0.96 (1.59)	6 (19)	8 (19)	13 (30)	14 (34)
	Winter hourly max.	11.27	39		392	
Other studies	[43] Delhi ^Win^	22.60 (14.90)			484 (230)	
	[43] Delhi ^Sum^	3.71 (4.29)			53,9 (136)	
	[12] Stockholm ^Spr,Aut,Win^	2.07 (1.62)				
	[24] Brisbane ^Spr,Aut,Win^	0.60				
	[31] Birmingham ^Sum,Aut^	1.30 (2.20)	23 (50)			
	[17] Oxford ^year^		15			
	[18] Southhampton ^year^		5			
	[46] Birmingham		17			55
	[42] Milan ^Spr,Aut,Win^		22–37		15–42	
	[45] Pekin ^Aut^				102.5	
	[15] Göteborg ^Aut,Win^			5		

For the “other studies”, the season is specified when it is clearly part of the campaign design. ^Spr^, ^Sum^, ^Aut^, and ^Win^ refer to spring, summer, autumn, and winter, respectively; ^year^ is used when the whole year was covered.

**Table 3 toxics-10-00033-t003:** PE mean for each environment and for each campaign (for BC, NO_2_, and PM_2.5_ for all volunteers).

	Environment	BC (ng·m^−3^)	NO_2_ (ppb)	PM_2.5_ (µg·m^−3^)
Spring	Commuting	3722 (1886)	24 (6)	6 (1)
	Polluted indoor	1555 (803)	11 (3)	115 (80)
	Indoor	574 (13)	9 (2)	5 (1)
	Outdoor	1986 (960)	19 (6)	6 (3)
Autumn	Commuting	4590 (932)	21 (7)	35 (6)
	Polluted indoor	3607 (1275)	11 (6)	113 (98)
	Indoor	1331 (115)	8 (3)	17 (2)
	Outdoor	3178 (379)	22 (7)	41 (6)
Winter	Commuting	2455 (782)	16 (4)	11 (8)
	Polluted indoor	1937 (1377)	11 (3)	53 (47)
	Indoor	542 (539)	5 (2)	8 (7)
	Outdoor	1187 (874)	11 (7)	15 (17)
	Wood burning	2085 (2104)	4 (2)	50 (88)

**Table 4 toxics-10-00033-t004:** Comprehensive correlations between PE to each pollutant (BC, NO_2_, and PM_2.5_) and background stations (Paris XIII^e^ for BC; Paris VII^e^, Paris XII^e^, and Paris XIII^e^ for NO_2_; and Paris IV^e^ for PM_2.5_) for each campaign (spring, autumn, and winter), for each time averaging (1-h and 1-day), for the correlations of Spearman and Pearson, and for each campaign as a whole (rs and rp) and in-between the volunteers (In-vol rs and In-vol r_P_). This last metric refers to the correlation between each volunteer (taken separately) and the monitoring station. The value range is given here (as min, max).

Campaign	Station	Pollutant	1-h Average				1-Day Average			
			r_S_	r_P_	In-vol. r_S_	In-vol. r_P_	r_S_	r_P_	In-vol. r_S_	In-vol. r_P_
Spring	Paris XIII^e^	BC	0.58	0.34	0.45, 0.74	0.23, 0.52	0.52	0.57	0.30, 0.90	0.29, 0.97
	Paris VII^e^	NO_2_	0.27	0.26	0.21, 0.47	0.14, 0.38	0.17	0.26	−0.70, 0.60	−0.49, 0.86
	Paris XII^e^	NO_2_	0.13	0.14	0.07, 0.47	0.05, 0.40	0.25	0.28	−0.30, 1.00	−0.24, 0.99
	Paris XIII^e^	NO_2_	0.19	0.24	0.07, 0.50	0.10, 0.51	0.25	0.30	−0.40, 1.00	−0.49, 1.00
	Paris IV^e^	PM_2.5_	0.51	0.12	0.15, 0.69	−0.14, 0.68	0.64	0.41	−0.20, 1.00	−0.51, 0.97
Autumn	Paris XIII^e^	BC	0.71	0.54	0.67, 0.79	0.51, 0.66	0.74	0.68	0.48, 0.95	0.49, 0.96
	Paris VII^e^	NO_2_	0.20	0.07	0.02, 0.52	−0.15, 0.49	0.07	0.02	−0.38, 0.90	−0.33, 0.93
	Paris XII^e^	NO_2_	0.13	0.10	−0.16, 0.55	−0.13, 0.64	0.10	0.17	−0.94, 0.67	−0.77, 0.74
	Paris XIII^e^	NO_2_	0.15	0.06	−0.07, 0.38	−0.18, 0.45	0.10	0.04	−0.83, 0.62	−0.83, 0.60
	Paris IV^e^	PM_2.5_	0.58	0.30	0.15, 0.88	0.00, 0.76	0.56	0.36	0.48, 0.90	−0.36, 0.94
Winter	Paris XIII^e^	BC	0.59	0.35	0.28, 0.77	0.07, 0.56	0.70	0.68	0.29, 0.93	0.07, 0.92
	Paris VII^e^	NO_2_	0.16	0.23	−0.08, 0.48	−0.01, 0.53	0.35	0.40	0.10, 0.80	0.18, 0.67
	Paris XII^e^	NO_2_	0.07	0.15	−0.17, 0.43	−0.15, 0.42	0.30	0.40	−0.40, 0.71	−0.05, 0.69
	Paris XIII^e^	NO_2_	0.10	0.19	−0.07, 0.38	−0.04, 0.43	0.35	0.41	−0.40, 0.79	−0.20, 0.65
	Paris IV^e^	PM_2.5_	0.66	0.23	0.39, 0.84	−0.01, 0.79	0.55	0.38	−1.00, 0.97	−1.00, 0.95

## Data Availability

The data is available upon written request to the corresponding author.

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
