# Peer review of "Personal Exposure to Black Carbon, Particulate Matter and Nitrogen Dioxide in the Paris Region Measured by Portable Sensors Worn by Volunteers"

_toxics, 2022, doi:10.3390/toxics10010033_

Round 1
Reviewer 1 Report
Poor indoor air quality has a negative effect on occupants’ health, comfort and performance. The importance of indoor air quality is addressed by present standards, which also provides the recommended measurement method. Determining the actual personal exposure to pollutants is not easy due to the phenomena occurring in the immediate human environment, in particular because of complicated air flow interactions in the breathing zone. The concentration of a given pollutant measured even at a short distance from a person may significantly differ from the concentration in the air introduced into the lungs. Current research indicates that the best solution for the assessment of exposure is the use of portable sensors worn by people [1, 2].
The article is very interesting. The authors do not hide the weaknesses of the research and explain them thoroughly, for example, the authors carefully comment on the effects of changes in the set of measuring devices or the impact of the volatility of daily habits of volunteers.
I find the attempt to correlate the precise indications of stationary devices monitoring the content of pollutants in the outside air with the indications of portable meters worn by people moving both in the outdoor and indoor environment as very valuable.
I have a few comments to the content of the presented article. The order of the comments does not reflect their significance. It results only from the order of appearance in the text of the manuscript:
- Lines 41-42, “It is also possible to combine pollution sensors with respiratory frequency and flow measurements to calculate exposure as the amount of pollutants actually inhaled [11 - 13].” - the results of tests in which sampling with respiratory phases was synchronized have recently been described in the literature. I suggest extending the introduction to include such information.
- Line 43, “from poor temporal resolution” - the issue of the required dynamics of indications of systems in exposure measurements is already recognized; I think it is worth mentioning.[3]
- Lines 124-125, “the expected ambient levels.” - needs clarification - what levels?
- Line 161, “hash” – typo? Maybe it was supposed to be "harsh".
- Line 167, “were invalidated” - Was it maybe some T-Student test? If so, please provide the confidence level.
- Line 185, “This must be taken into account when comparing the different campaign results” - Why were the same volunteers not involved in the study, or other volunteers who move to the same locations?
- Figure 2, - where is outdoor ?
- Line 262, “Some volunteers” - The way the measurements were carried out seems to be rather spontaneous, the number of measures changes, the type of participants changes, the procedure changes. Or rather, one should not perform such tests more systematically and thus allow for a reliable comparison of the results.
- Line 399, “lighter” - This term somehow does not fit, I thought about the word "hesitant".
- Line 425, “from the output of a predictive model” - This should be commented on more widely; Which model ?; What assumptions will the calculations be based on? What would be the accuracy of such a model?
- Ghahramani A, Pantelic J, Lindberg C, Mehl M, Srinivasan K, Gilligan B, et al. Learning occupants' workplace interactions from wearable and stationary ambient sensing systems. Applied Energy. 2018 Nov;230:42-51.
- Kierat W, Melikov A, Popiolek Z. A reliable method for the assessment of occupants' exposure to CO2. Measurement. 2020 Oct;163.
- Kierat W, Bivolarova M, Zavrl E, Popiolek Z, Melikov A. Accurate assessment of exposure using tracer gas measurements. Building and Environment. 2018 MAR 2018;131:163-73.
Reviewer 2 Report
- General appreciation
This manuscript addresses a very interesting topic – assessing personal human exposure to air pollutants in a very populated region of Paris, France. This assessing is based on portable monitoring sensors worn by people, which offer many advantages but also present important drawbacks. In general, the manuscript is also well written and organized, although has some gaps and unclear parts. So, some revisions are required before it can be considered acceptable for publishing.
2. Major Concerns
The main weakness of this manuscript is related with the uncertainties of the results. The authors are awareness of this weak point, but in my opinion, the methodological approach did not value certain important aspects that could help in increasing the reliability of the data. The following questions stress some of these critical points for which I would appreciate the answers/comments from the authors:
- why did the authors not carry out pedestrian control circuits, covering the different microenvironments studied, in similar proportions in terms of exposure time, and having at least 3 volunteers walk through it simultaneously? This action would help to understand if the differences found reflected effectively the real spatial and temporal patterns or they were simply caused by the (apparently) low reliability of the sensors used in the study.
- Taking into account some information shared in the manuscript, most volunteers spent most time indoors, where they also were exposed to high pollution levels. So, what is the scientific relevance of evaluating the effectiveness of the stationary air quality monitoring stations as PE surrogate by using the data collected in this study?
- General and some specific comments
Introduction:
The introduction is poor in some topics, especially with regard to the type of equipment used, which have many advantages over others, but also many disadvantages associated to its nature and the measurement process itself. So, it should be improved.
Material and Methods
This chapter remits an important part, related to the validation and selection of used sensors, to another article, but it is still unclear whether the sensors tested were the ones used or if new sensors were purchased. Both situations may require further validation tests. According to the authors the validation tests were performed in 2017. Authors must add some comments about this point that in my opinion and based in my experience is critical. Furthermore, this chapter has some unclear information: how many devices of the same parameters were used by each volunteer? how many days were evaluated? how many volunteers participated in the study (in the introduction they mention 139!!!)? how many days of sampling did each volunteer participate?
Results
line 180 - I don't think there is as much similarity in spatial representativeness for spring and autumn as the authors refer. How much can this affect the results in terms of PE seasonality?
line 195 - what kind of seasonal variations the authors are talking about? emissions sources? meteorological and/or indoor ventilation conditions? behaviour patterns of people (volunteers)?
Line 202 - how did the authors come to this conclusion? The authors must explain?
Line 252 - which indoor pollution episode do are the authors referring to?
line 257 - weren't the volunteers supposed to register anomalous situations?
Figure 2- Why BC values ​​are greater than PM, particularly in the periods designated as "Polluted indoor". Have the authors any plausible explanation?
lines 262-284 - The volunteer AH spent all his daily time indoors. So, it would be very important to show data for volunteers with exposure periods to ambient air pollutants for many reasons. Furthermore, it will be useful to provide more detailed information about indoor, outdoor, commuting, etc. environments as they are or can be so different.
Figure 5 - the authors have to explain how they calculate the normalized level. Even after doing that, It seems me difficult to infer valuable information from the different curves. Maybe authors can explain better and in a consistent way the information provided by the figure.
Lines 351-365 and 400-406 - The comparative analysis is focused only on magnitudes of correlation coefficients. This is not enough neither the most important way to validate the obtained results.
Table 4 - As exposed before, the analysis presented in the table doesn't make sense in my opinion. What make sense is to compare outdoor concentrations measured by the portable sensors worn by volunteers with the concentrations provided by the stationary ambient air monitoring stations.
Still in relation to this table, authors must explain better the meaning of in-between the volunteers.
Discussion and Conclusion
There is little or no relevant discussion. It should be improved.
Round 2
Reviewer 2 Report
I was minimally satisfied with the answers to my questions and with the changes made in the manuscript, which is now less opaque and reinforce in a clearer way the need to continue to develop studies of this nature and continue to improve low-cost measurement systems to be part of our future.